# Chemical and Biological Evaluation of Amazonian Medicinal Plant *Vouacapoua americana* Aubl

**DOI:** 10.3390/plants12010099

**Published:** 2022-12-25

**Authors:** Serhat Sezai Çiçek, Anna Laís Pfeifer Barbosa, Arlette Wenzel-Storjohann, Jorge Federico Orellana Segovia, Roberto Messias Bezerra, Frank Sönnichsen, Christian Zidorn, Isamu Kanzaki, Deniz Tasdemir

**Affiliations:** 1Department of Pharmaceutical Biology, Kiel University, Gutenbergstraße 76, 24118 Kiel, Germany; 2Research Unit Marine Natural Products Chemistry, GEOMAR Centre for Marine Biotechnology (GEOMAR-Biotech), GEOMAR Helmholtz Centre for Ocean Research Kiel, Am Kiel-Kanal 44, 24106 Kiel, Germany; 3Ecoregional Research Unit, Brazilian Agricultural Research Corporation, Rod. JK, Km 5, Macapá 68903-419, AP, Brazil; 4Laboratory of Bioprospection and Atomic Absorption, Federal University of Amapá, Rod. JK, Macapá 68903-419, AP, Brazil; 5Otto Diels Institute for Organic Chemistry, Kiel University, Otto-Hahn-Platz 4, 24118 Kiel, Germany; 6Laboratory of Bioprospection, Darcy Ribeiro Campus, University of Brasilia, Brasilia 70910-900, DF, Brazil; 7Faculty of Mathematics and Natural Sciences, Kiel University, Christian-Albrechts-Platz 4, 24118 Kiel, Germany

**Keywords:** Caesalpinioideae, acapu, vouacapenic acid, methyl vouacapenate, MRSA, *E. faecium*, natural product, antimicrobial, cytotoxic, antibacterial

## Abstract

*Vouacapoua americana* (Fabaceae) is an economically important tree in the Amazon region and used for its highly resistant heartwood as well as for medicinal purposes. Despite its frequent use, phytochemical investigations have been limited and rather focused on ecological properties than on its pharmacological potential. In this study, we investigated the phytochemistry and bioactivity of *V. americana* stem bark extract and its constituents to identify eventual lead structures for further drug development. Applying hydrodistillation and subsequent GC-MS analysis, we investigated the composition of the essential oil and identified the 15 most abundant components. Moreover, the diterpenoids deacetylchagresnone (**1**), cassa-13(14),15-dien-oic acid (**2**), isoneocaesalpin H (**3**), (+)-vouacapenic acid (**4**), and (+)-methyl vouacapenate (**5**) were isolated from the stem bark, with compounds **2** and **4** showing pronounced effects on Methicillin-resistant *Staphylococcus aureus* and *Enterococcus faecium*, respectively. During the structure elucidation of deacetylchagresnone (**1**), which was isolated from a natural source for the first time, we detected inconsistencies regarding the configuration of the cyclopropane ring. Thus, the structure was revised for both deacetylchagresnone (**1**) and the previously isolated chagresnone. Following our works on *Copaifera reticulata* and *Vatairea guianensis*, the results of this study further contribute to the knowledge of Amazonian medicinal plants.

## 1. Introduction

*Vouacapoua americana* Aubl. (Fabaceae—Caesalpinioideae) is one of four species of the genus *Vouacapoua* and is distributed throughout the Amazon region, including Brazil, Guyana, French Guiana, Suriname, and Venezuela, where it is referred to as acapu (Brazil) or wacapou (French Guiana, Suriname) [1,2,3]. It is a tree of up to 40 m in height with a stem diameter of up to 100 cm and is appreciated for its durable and highly resistant heartwood, which makes it an important source for all types of heavy-duty constructions [2,4]. The plant is also used for medicinal purposes; decoctions of the stem bark are applied as an astringent, antiseptic, and fortifier and are used for the treatment of diarrhea, gastric ulcers, hemorrhoids, spinal pain, and ailments of the liver [3,5,6].

Despite the ethnomedical usage of *V. americana*, bioactivity studies so far have only focused on the species’ phytotoxic and fungitoxic potential [7,8]. Thereby, aqueous extracts of the stem bark showed allelopathic activity, inhibiting germination and primary root elongation of pasture weeds *Mimosa pudica* and *Urena lobata* [7]. Since a follow-up study found the aqueous extract to contain no organic compounds but high concentrations of inorganic ions, the allelopathic effects were attributed to saline stress [8]. Similarly, investigations on the phytochemistry of *V. americana* have been scarce, with only three reported chemical structures so far [9,10,11]. Spoelstra [9] reported the presence of sesquiterpenes and sesquiterpene alcohols, as well as one crystalline substance in the benzene extract of *V. americana* bark, which was later identified as (+)-methyl vouacapenate (**5**) together with the corresponding free acid (**4**) (Figure 1) [10]. Another diterpenoid was reported by Kido et al. [11], who isolated cassa-13(14),15-dien-oic acid (**2**) together with the two aforementioned compounds.

As part of our ongoing research on Amazonian medicinal plants [12,13,14,15], we investigated the phytochemical profile of *V. americana* stem bark leading to the identification of the essential oil composition and the isolation of five diterpenoids (**1**–**5**, Figure 1), of which **1** was isolated from a natural source for the first time. In addition, we evaluated the isolated metabolites for their bioactivity potential towards a small panel of antimicrobial and cytotoxicity assays.

## 2. Results and Discussion

### 2.1. Essential Oil Composition

In the first step, the essential oil composition of *V. americana* stem bark was evaluated. Therefore, 5 g of dry plant material was ground and hydrodistilled by the method of the European Pharmacopeia [16]. The distillate was subsequently analyzed by GC-MS, using the Adams method [17] and a comparison of retention indices. Additionally, MS spectra were matched with spectra from the Adams and NIST databases. All compounds showing a relative content of more than 0.2%, a deviation of retention indices of less than 10, and (reversed) similarity indices of more than 700 are listed in Table 1.

Altogether, 14 sesquiterpenes and one diterpene ester (**5**) were found in the respective concentration and could be unambiguously identified. Interestingly, the diterpenoid (**5**) amounted to three-quarters of the distillate, whereas the sum of sesquiterpenes was below 25%. This ratio is in accordance with the results obtained by Spoelstra [9], who found a benzene extract to contain approximately 1.5% of volatiles and 3–3.5% of a crystalline substance after fractionation with vacuum distillation or high vacuum distillation, respectively.

In addition, Spoelstra reported the presence of cadinene and sesquiterpene alcohols, which is also supported by our results and which we could further specify. Of the volatile fraction, δ-cadinene was found to be the most dominant sesquiterpene with a concentration of 8.3%, followed by α-calacorene and α-muurolene as well as sesquiterpene alcohols τ-muurolol and α-cadinol, all showing amounts between 3 and 4%. Other sesquiterpenes with noteworthy concentrations were β-caryophyllene, γ-muurolene, and *trans*-cadina-1,4-diene, with concentrations of more than 2%.

### 2.2. Isolation and Structure Elucidation

In the next step, *V. americana* stem bark was extracted with methanol on a preparative scale (980 g of plant material, 6 L of solvent). The crude extract was further separated by liquid-liquid-extraction, column chromatography, and semi-preparative HPLC to afford five compounds, which were investigated by mass spectrometry and NMR spectroscopy and for their optical rotation values. After comparison of the obtained data with the values reported in the literature, compounds **2**, **4,** and **5** were identified as cassa-13(14),15-dien-oic acid (**2**), (+)-vouacapenic acid (**4**), and (+)-methyl vouacapenate (**5**), which were previously reported for *V. americana* [9,10,11]. Compound **3** was not described for *V. americana* but has been isolated from tara (*Caesalpinia spinosa*) and named isoneocaesalpin H [18].

Compound **1** (deacetylchagresnone) was isolated as a white powder, and its molecular formula was deduced as C_20_H_32_O_3_ based on HRESIMS measurements (Appendix A). Analysis of the NMR data (Table 2, Appendix A) indicated 20 carbon resonances, corresponding to three methyl groups (*δ*_C_ 27.0, 14.8, and 14.1), eight methylene groups (*δ*_C_ 65.2, 62.4, 38.1, 36.5, 35.5 (2C), 22.1, and 18.2), five methine groups (*δ*_C_ 57.0, 55.2, 38.3, 37.2, and 36.8), three quaternary carbons (*δ*_C_ 38.5, 37.2, and 33.5), and one carbonyl carbon (*δ*_C_ 211.4).

In positions 1–8, carbon shift values of compound **1** were similar to those of the other isolated compounds (Appendix A), thus also suggesting a diterpene scaffold. However, except for the carbonyl group, the NMR data of **1** neither showed alkene double bonds nor a carboxyl group in position 4, as observed for the other isolated constituents (Figure 1, Appendix A). Instead, compound **1** was substituted with a hydroxymethylene group in this position as indicated by the characteristic shift values of 65 (^13^C) and 3.4 to 3.7 (^1^H) ppm, respectively, as well as HMBC correlations to C-3, C-4, and C-18. NOESY correlations to H-2 and CH_3_-20, furthermore, confirmed the hydroxymethylene group to be in β-position (Figure 2).

HMBC couplings of the carbonyl group to both H-11 protons and two methine groups indicated position 12 for this feature. More interestingly, the carbonyl group was also coupling to the protons of the CH_3_-17 methyl group, thus suggesting the diterpenoid to be of a pimarane-type in contrast to the before observed cassane-type diterpenes. This was corroborated by the missing signal splitting of CH_3_-17. The corresponding ethyl group was found to be hydroxylated in the terminal position and, moreover, to be linked to the C-14 carbon, thus forming a cyclopropane ring.

A literature search for corresponding compounds yielded the hydrolyzation product of chagresnone, which was previously isolated from *Myrospermum frutescens* (Fabaceae) [19]. The NMR data of compound **1** matched perfectly with the reported values. However, the proposed α-orientation of the cyclopropyl group, which was not based on respective NOESY correlations, was not supported by our experiments. For example, no correlations were observed between H-8 and neighboring H-14 or CH_3_-17, respectively. In contrast, NOESY correlations supported the β-orientation of the cyclopropyl group, as demonstrated by the coupling of H-15 with H-8 and H-11a or the coupling of H-14 with H-7b, respectively (Figure 2). Therefore, the cyclopropyl group had to be β-oriented, and H-14, as well as CH_3_-17, had to be α-oriented instead, as was the CH_2_-16 methylene group. Thus, the structure of compound **1** was established as (14R,16R,18S)-16,18-dihydroxy-14,15-cyclopimaran-12-one.

### 2.3. Bioactivity Assessments

All isolated compounds were subjected to a panel of antimicrobial and cytotoxicity assays (Appendix A). Studies on cytotoxicity comprised of the following cell lines: breast cancer (MB-231), colon cancer (HCT-116), lung cancer (A-549), melanoma (A-375), and non-cancerous keratinocytes (HaCaT). Antimicrobial activity assessments were carried out on human pathogenic yeast/fungi (*Candida albicans, Cryptococcus neoformans*, *Trichophytum rubrum*) as well as on the ESKAPE panel comprised of highly virulent human pathogenic bacteria. This included the gram-positive bacteria Methicillin-resistant *Staphylococcus aureus* (MRSA), *Enterococcus faecium,* and the gram-negative bacteria *Acinetobacter baumanii, Escherichia coli, Klebsiella pneumoniae, Pseudomonas aeruginosa*.

At a test concentration of 100 µM, none of the five compounds exhibited noteworthy cytotoxicity (Appendix A). The highest effect was observed for compound **5**, which showed 60% growth inhibition against both A-375 and HaCaT cell lines. The activity of **5** against other cell lines and of other compounds was significantly below 50%. The same picture was observed against fungi and gram-negative bacteria, with inhibition rates below 50% and, thus, IC_50_ values above 100 µM (Appendix A). However, two compounds were found to be active against gram-positive bacteria (Appendix A). Compounds **2** and **4** inhibited the growth of MRSA with IC_50_ values of 13.5 µM and 12.0 µM, respectively (Table 3). (+)-Vouacapenic acid (**4**) also displayed activity against *E. faecium*, with an IC_50_ value of 8.3 µM, thus being active against both gram-positive test strains.

The pronounced activity of some diterpene acids is well known, and phytochemical research on resins and other sources of this particular compound class, therefore, is still relevant. Caesalsappanine J, which was isolated from *Caesalpinia sappan*, shows strong structural similarities to compound **3** and exhibited antiproliferative activity against the KB cell line with an IC_50_ value of 7.4 µM [20]. In addition, caesalsappanins O, P, and Q displayed moderate effects against MFC-7 and HCT-116 cells [21]. Other reported cytotoxic diterpenoids from *C. sappan* are caesappins A and B, with weak to moderate effects against HeLa and HepG-2 cells [22]. Cassane-type diterpenoids with more pronounced effects were isolated from *Erythrophleum fordii* (erythroformin A and B) and *Euphorbia fischeriana* (euphkanoid H) [23,24]. While erythroformins A and B affected lung cancer cell lines A-549, NCI-H1975, and NCI-H1229 with IC_50_ values in the range of 0.4 to 1.4 µM [23], euphkanoid H showed activity on HEL cells with an IC_50_ value of 3.2 µM [24]. However, in contrast to the diterpenoids isolated in the present study, the latter three compounds bore a nitrogen atom. *E. fischeriana*, furthermore, yielded a series of moderately (IC_50_ values of 10–20 µM) cytotoxic *ent*-abietane-type diterpenoids, such as euphonoids A–D with activity against C4-2B cells [25], fischerianoids A and B with effects on MM-231 and Hep3B cells [26], or difischenoid A showing growth inhibition of HeLa cells [27]. The latter study also yielded difischenoid B, which exhibited stronger effects on HeLa cells (IC_50_ value of 3.75 µM) and was, moreover, active on MCF-7 cells (IC_50_ value of 9.31 µM). Phytochemical investigation of the Chinese mangrove *Ceriops tagal* yielded the cytotoxic pimarane-type diterpene tagalon C and the isopimarane-type diterpene tagalon D [28]. Both compounds were effective against the breast cancer cell line MT-1, with IC_50_ values of 3.75 µM and 8.07 µM, respectively.

Also, with regard to antimicrobial activities, compounds of various diterpene types were found effective. Isopimaric acid and abietic acid, for example, were found to inhibit the growth of MRSA with MIC values of 32 to 64 µg/mL, while carnosic acid was found to modulate drug resistance via the inhibition of efflux pumps [29,30]. In a previous study on the oleoresin of *Copaifera reticulata*, we identified five compounds of three different diterpene types with significant activity against both MRSA and *E. faecium* with IC_50_ values of 10.7 to 2.5 µM and 9.3 to 1.6 µM, respectively [13]. Thereby, the activity increased with the lipophilicity of the tested components. (–)-Polyalthic acid, the major diterpenoid in *C. reticulata* and the only isolated diterpene bearing a furyl moiety, was also effective against *Trichophytum rubrum*, *T. mentagrophytes*, and *Cryptococcus neoformans* [14]. In contrast to the antifungal activity, the antibacterial effects completely vanished upon esterification or amidation, a fact that was also observed in the present study. While (+)-vouacapenic acid (**4**) showed pronounced activity on both gram-negative bacteria, its corresponding methyl ester (**5**) did not exhibit any effects at all (Table 3). However, in the case of isoneocaesalpin H (**3**), antibacterial effects are missing, despite the structural similarity to (+)-vouacapenic acid (**4**) and a carboxyl group in position 4 (Figure 1). If the missing activity is also due to higher polarity (caused by one additional hydroxy group in position and one oxo group in position (16) cannot be excluded and is at least corroborated by the activity of the more lipophilic but less similar compound **2**. The same question is even more difficult to answer for deacetylchagresnone (**1**), which shows higher polarity than the active compounds as well as a hydroxymethylene group instead of a carboxylic acid in position 4. Moreover, compound **1** displays a different diterpene type, namely a pimarane derivative. Therefore, additional derivatives must be isolated in order to have sufficient data for detailed structure-activity-relationship studies. Hence, the discovery of deacetylchagresnone (**1**), which has not been isolated from nature before, and the identification of two potent antibacterials, render *V. americana* an interesting source for the discovery of further bioactives. The activity against *E. faecium*, one of the major pathogens for nosocomial infections, is of particular interest, as comparably few inhibiting natural products have been identified so far, while vancomycin resistances continue to decrease [31].

## 3. Materials and Methods

### 3.1. General Experimental Procedures

Thin layer chromatography was performed with precoated TLC plates (Silica gel 60 F254 Merck, Darmstadt, Germany) using dichloromethane–methanol (19:1) as eluent and vanillin sulfuric acid as spraying reagent. Flash chromatography was carried out manually with a glass column (500 × 235 mm) or with a Büchi PrepChrom C-700 chromatograph using a FlashPure EcoFlex Silica Gel SL cartridge (40 g, irregular 40–63 µM particle size, Büchi Labortechnik GmbH, Essen, Germany). Preparative MPLC was conducted on a Büchi PrepChrom with a Büchi PrepChrom HPLC column (C18, 15 µM, 250 × 30.0 mm), and size exclusion chromatography was performed with Sephadex LH-20 (GE Healthcare AB, Uppsala, Sweden). HPLC-DAD-ELSD analyses were accomplished on a VWR-Hitachi Chromaster Ultra RS equipped with a 6170 binary pump, 6270 autosampler, 6310 column oven, 6430 diode array detector, and a VWR 100 evaporative light scattering detector (VWR International GmbH, Darmstadt, Germany). UHPLC-MS analyses were carried out on a Shimadzu Nexera 2 liquid chromatograph connected to an LCMS 8030 triple quadrupole mass spectrometer using electrospray ionization (Shimadzu, Kyoto, Japan). A Phenomenex Luna Omega C18 column (100 × 2.1 mm, 1.6 µM particle size, Phenomenex, Aschaffenburg, Germany) was employed for the analysis of extracts, fractions, and pure compounds with 0.1% formic acid in water (A) and acetonitrile (B) and the following gradient: 35% B to 95% B in 30 min, to 95% B in 15 min. Post run: 10 min. Flow rate: 0.2 mL/min, column temperature: 35 °C. Hydrodistillation was accomplished as described in the European Pharmacopeia (10th edition) using 5 g of plant material and 200 mL of water. Distillation was running for 3 h with a rate of 3 mL/min. 500 µL of xylene was used to collect the essential oil, which was diluted to 1 mL with xylene and injected into the GC-MS/MS apparatus with a split of 1:10. Analyses were performed on a Trace 1310 gas chromatograph equipped with an SSL and PTV inlet and a TSQ Duo mass spectrometer and a TG-5SilMS (30 m × 0.25 mm × 0.25 µm) (Thermo Scientific, Bremen, Germany). The following parameters were applied: 60 °C, 3 °C/min to 246 °C, and hold for 8 min. MS parameters: Fullscan 50–500 m/z. Compounds were identified using the NIST database version 2020 and the Adams enhanced database (4th edition), as well as a comparison of retention indices. HRESIMS spectra were recorded on a Q-Exactive Plus spectrometer, and specific rotation of the compounds was measured on a Jasco P-2000 polarimeter (Jasco, Pfungstadt, Germany). 1D (^1^H, ^13^C) and 2D (HSQC, HMBC, COSY, NOESY) NMR spectra were recorded on a Bruker Avance III 400 NMR spectrometer operating at 400 MHz for the proton channel and 100 MHz for the ^13^C channel with a 5 mm PABBO broadband probe with a z gradient unit at 293 K (Bruker BioSpin GmbH, Rheinstetten, Germany). A Bruker Avance 600 MHz spectrometer was used for measurements of compound **1**. Reference shift values were 7.26 (^1^H) and 77.36 (^13^C) for chloroform, as well as 3.31 (^1^H) and 49.15 ppm (^13^C) for methanol, respectively. Conventional 5 mm NMR sample tubes were obtained from Rototec-Spintec GmbH, Griesheim, Germany. Alkane standard (C7–C40) and LC-MS grade formic acid were purchased from Sigma Aldrich Co., St. Louis, MO, USA. LC-MS grade acetonitrile and water, gradient grade methanol, and other (analytical grade) solvents were obtained from VWR International GmbH, Darmstadt, Germany. Water used for isolation was doubly distilled in-house. Chloroform-*d*_1_ (99.50%, Lot T2811, Batch 0920) and methanol-*d*_4_ (99.80%, Lot P3021, Batch 1016B) for NMR spectroscopy were purchased from Euriso-top GmbH, Saarbrücken, Germany.

### 3.2. Plant Material

Stem bark of *Vouacapoua americana* was collected in February 2020 in the municipality of Serra do Navio, Amapá, at the coordinates 00°53′20.9″ N/052°00′06.5″ W and identified by J.F.O. Segovia. A voucher specimen is kept in the Dr. J.F.O. Segovia collection of the herbarium of the Brazilian Agricultural Research Corporation (EMBRAPA) in Macapá, under the registration number ESK-03. The plant species was registered for access to the genetic heritage in SisGen with process no. 00.038.174/0001-43.

### 3.3. Extraction and Isolation

980 g of dried and ground stem bark were extracted five times with 1.2 L of methanol using ultra-sonication for 15 min and subsequent maceration for at least 12 h. The solvent was evaporated under reduced pressure at a temperature of 35 °C to afford 237 g of crude extract, which was then partitioned between water and dichloromethane. The dichloromethane fraction (27.0 g) was subjected to column chromatography (500 × 235 mm) using silica gel (800 g) as the stationary phase, and the following gradient of *n*-hexane–dichloromethane–methanol: 75:25:0 (2 L), 50:50:0 (2 L), 25:75:0 (2 L), 0:100:0 (2 L), 0:98:2 (2 L), 0:96:4 (2 L), 0:94:6 (1 L), 0:92:8 (1 L), 0:90:10 (1 L), and 0:0:100 (2 L). Of the resulting 14 fractions, fraction 5 yielded 2.22 g of compound **5** after crystallization from acetone. Fraction 7 yielded 2.40 g of compound **2,** and fraction 10 gave 1.88 g of compound **4** after crystallization from acetone. Fractions 12 (233 mg) and 13 (984 mg) were combined and subjected to silica gel flash chromatography with the following gradient of *n*-hexane–ethyl acetate–methanol: 100:0:0 in 10 min to 50:50:0, in 10 min to 40:40:20, in 10 min to 0:0:100, in 30 min to 0:0:100. Of the resulting five fractions, the latter fraction (289 mg) was separated by Sephadex LH-20 column chromatography [acetone-methanol (1:1)]. Fraction 4 (out of 7) was further separated by semi-preparative HPLC to afford 9.22 mg of compound **1** and 13.8 mg of compound **3**.

*Deacetylchagresnone* (**1**): White powder; m.p. 80.2 °C; [α]^25^_D_ + 199 (c 2.85, methanol), HRESIMS (positive) *m/z* 353.2658 [M + MeOH + H]^+^ (calcd for C_21_H_37_O_4_, 353.2692) (Appendix A); ^1^H NMR (CDCl_3_, 600 MHz) and ^13^C NMR (CDCl_3_, 150 MHz) see Table 2 and Appendix A.

### 3.4. Biological Activity

Antibacterial and anti-yeast activity was determined as described by Çiçek et al. [14]. The bacteria were cultivated in TSB medium (1.2% tryptic soy broth, 0.5% NaCl), except *E. faecium,* which was cultivated in M92 medium (3% trypticase soy broth, 0.3% yeast extract, pH 7.0–7.2). The cultivation of *C. albicans* and *C. neoformans* took place in M186/3 (0.3% glucose, 0.17% peptone from soymeal, 0.1% malt extract, 0.1% yeast extract). Overnight cultures of the test organisms were prepared and diluted to an optical density (600 nm) of 0.01–0.03. To prepare the assay, the test samples (20 mg/mL DMSO stock solution) were dissolved in medium and transferred into a 96-well microtiter plate, and 200 µL of the cell suspension cultures were added to each well. The final assay concentration of the substances was 100 µg/mL, and the final DMSO concentration was 0.5% (negative control). For IC_50_ determination, a dilution series was prepared, and the IC_50_ value was calculated by Excel or GraphPad Prism as the concentration that shows 50% inhibition of viability on the basis of a negative control.

Activity assays against dermatophytic fungi and cytotoxicity assays were performed as described by Pfeifer Barbosa et al. [13]. *Trichophyton rubrum* I/95 (patient isolates from Kiel University Hospital) were cultivated for two weeks on Sabouraud agar (1% peptone, 2% glucose, 1.5% agar-agar, pH 5.6). A suspension of 5 × 10^4^ spores/mL Sabouraud medium was prepared and added to each well of a microplate containing the test samples, which were transferred as described before (antibacterial assay). After incubation for 72 h at 28 °C and 120 rpm, the optical density at 600 nm was measured using the microplate reader. The resulting values were compared with a positive (clotrimazole) and a negative control (0.5% DMSO) on the same plate.

For cytotoxicity assays, HEP-G2, HT-29, and HaCaT cells were cultivated in RPMI medium, A549 and MDA-MB-231 cells in DMEM:Ham’s F12 medium (1:1) supplemented with 15 mM HEPES, and A-375 and HCT-116 cells were cultivated in DMEM medium supplemented with 4.5 g/L D-glucose and 110 mg/L sodium pyruvate. All media were supplemented with L-glutamine, 10% fetal bovine serum, 100 U/mL penicillin, and 100 µg/mL streptomycin. The cultures were maintained at 37 °C under a humidified atmosphere and 5% CO_2_. The cell lines were transferred every 3 or 4 days. For the experimental procedure, cells were seeded in 96-well plates at a concentration of 10,000 cells per well in RPMI. After 24 h incubation, the medium was removed, and 100 µL of the test sample was adjusted to a final concentration of 100 µg/mL by diluting with a growth medium. Doxorubicin, as a standard therapeutic drug, was used as a positive control, and DMSO (0.5%) was used as a negative control.

## 4. Conclusions

In the present study, we conducted a detailed phytochemical analysis of *Vouacapoua americana* stem bark extract, leading to the elucidation of the essential oil composition and the isolation of five diterpenoids. The isolated substances were subsequently tested in a panel of antimicrobial and cytotoxicity assays, revealing pronounced effects for cassa-13(14),15-dien-oic acid (**2**) and (+)-vouacapenic acid (**4**) against Methicillin-resistant *Staphylococcus aureus* and *Enterococcus faecium*, respectively. Additionally, one of the compounds was isolated from a natural source for the first time and identified as deacetylchagresnone (**1**). We detected inconsistencies in the structure elucidation of the related compound chagresnone leading to the wrong assignment of the cyclopropane orientation, which we clarified thanks to detailed NOESY correlations. Following our previous work on *Copaifera reticulata* and *Vatairea guianensis*, the present study is another contribution to the chemical composition and bioactivity of Amazonian medicinal plants.

## Figures and Tables

**Figure 1 plants-12-00099-f001:**
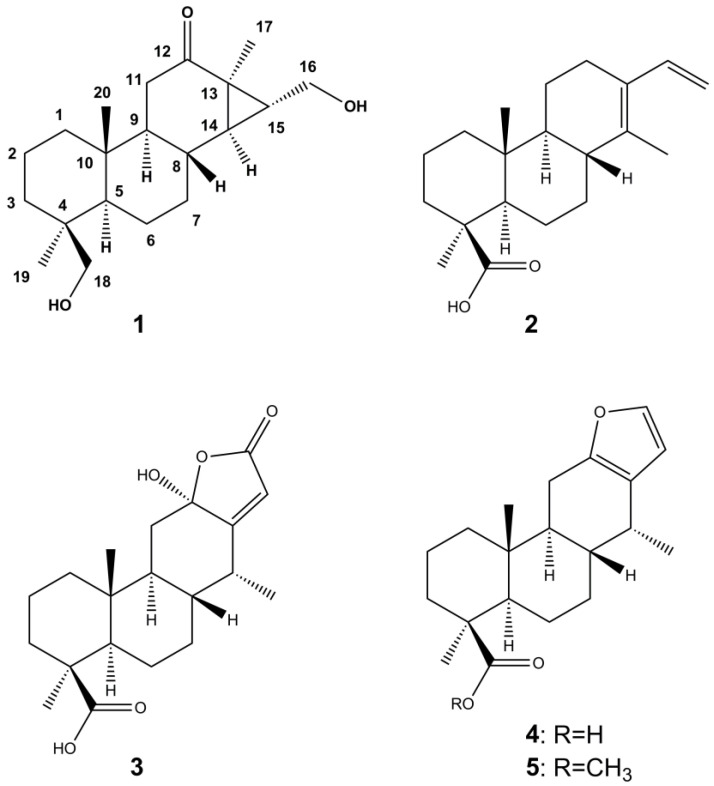
Chemical structures of isolated compounds deacetylchagresnone (**1**), cassa-13(14),15-dien-oic acid (**2**), isoneocaesalpin H (**3**), (+)-vouacapenic acid (**4**), and (+)-methyl vouacapenate (**5**).

**Figure 2 plants-12-00099-f002:**
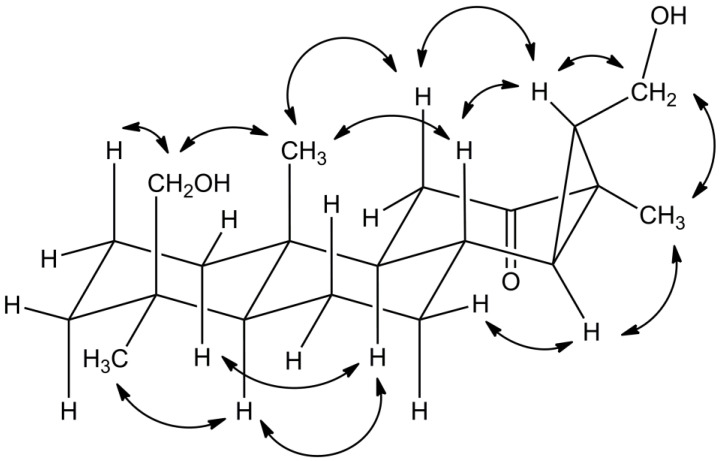
Important NOESY correlations observed for compound **1**.

**Table 1 plants-12-00099-t001:** Composition of the essential oil identified by GC-MS analysis and retention indices. (+)-methyl vouacapenate (**5**) was retrospectively identified by comparison with the isolated substance.

Compound Name	t_R_ ^1^ [min]	ToC ^2^ [%]	CoS ^3^ [%]	RI_exp_ ^4^	RI_lit_ ^5^	SI ^6^	RSI ^7^	ID ^8^
α-copaene	19.56	0.23	1.0	1375	1374	856	874	RI, MS
β-caryophyllene	21.30	0.63	2.7	1418	1417	906	910	RI, MS
rotundene	22.95	0.35	1.5	1459	1457	812	901	RI, MS
γ-muurolene	23.48	0.60	2.6	1473	1478	925	933	RI, MS
α-muurolene	24.42	0.71	3.0	1496	1500	924	931	RI, MS
γ-cadinene	24.96	0.31	1.3	1510	1513	894	911	RI, MS
δ-cadinene	25.21	1.94	8.3	1516	1522	872	900	RI, MS
*trans*-cadina-1,4-diene	25.71	0.53	2.3	1529	1533	894	943	RI, MS
α-calacorene	26.01	0.91	3.9	1537	1544	891	933	RI, MS
caryophyllene oxide	27.53	0.28	1.2	1577	1582	837	848	RI, MS
τ-muurolol	29.87	0.70	3.0	1640	1640	868	880	RI, MS
α-cadinol	30.31	0.69	3.0	1651	1652	856	865	RI, MS
cadalene	30.90	0.40	1.7	1668	1675	750	842	RI, MS
amorpha-4,9-dien-2-ol	31.88	0.27	1.2	1694	1700	771	785	RI, MS
(+)-methyl vouacapenate (**5**)	56.68	76.67		2446	—	—	—	Ref.

^1^ **t_R_**: Retention time on the TG-5SilMS GC column. ^2^**ToC**: Concentration relative to all compounds in the distillate given in percent. ^3^**CoS**: Concentration relative to sesquiterpenes found in the distillate given in percent. ^4^**RI_exp_**: Retention index determined relative to *n*-alkanes (C10-C25). ^5^ **RI_lit:_** Retention index reported by Adams [17]. ^6^ **SI**: Similarity index of mass spectra. ^7^ **RSI**: Reversed similarity index of mass spectra. ^8^
**ID**: Methods used for identification. RI: Comparison of Retention indices, MS: Comparison to MS databases, Ref.: Comparison with reference standard.

**Table 2 plants-12-00099-t002:** ^1^H (600 MHz) and ^13^C (150 MHz) NMR data of compound **1** in CDCl_3_ (δ in ppm, *J* in Hz).

Position	^1^H NMR	^13^C NMR
1	0.89 m, 1.60 dt (3.9, 12.7)	38.1, CH_2_
2	1.46 (2H) m	18.2, CH_2_
3	0.93 m, 1.80 m	35.5, CH_2_
4		38.5, C
5	1.03 dd (2.5, 12.7)	55.2, CH
6	1.34 m, 1.78 m	22.1, CH_2_
7	1.23 m, 2.05 m	35.5, CH_2_
8	1.67 m	36.8, CH
9	1.12 td (1.9, 11.7)	57.0, CH
10		37.2, C
11	1.85 t (14.1), 2.20 dd (2.1, 14.3)	36.5, CH_2_
12		211.4, C
13		33.5, C
14	1.01 dd (1.5, 12.9)	38.3, CH
15	1.51 m	37.2, CH
16	3.56 dd (7.8, 11.6), 3.79 dd (6.0, 11.7)	62.4, CH_2_
17	1.23 s	14.1, CH_3_
18	0.95 s	27.0, CH_3_
19	3.42 d (10.9), 3.72 d (10.9)	65.2, CH_2_
20	0.76 s	14.8, CH_3_

**Table 3 plants-12-00099-t003:** Antibacterial effects of compounds **1**–**5**. The IC_50_ values are in µM. Positive controls (PC) were chloramphenicol (MRSA) and ampicillin (*E. faecium*), respectively.

	1	2	3	4	5	PC
MRSA	–	13.5 ± 5.6	–	12.0 ± 5.1	–	4.6 ± 0.1
*E. faecium*	–	–	–	8.3 ± 1.3	–	0.6 ± 0.0

## Data Availability

Data are available from the authors upon request.

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
