# Peer review of "Chemical and Biological Evaluation of Amazonian Medicinal Plant Vouacapoua americana Aubl"

_plants, 2022, doi:10.3390/plants12010099_

Round 1

Reviewer 1 Report

This manuscript is within the proposed subject matter for the special issue of Plants and the English writing seems fine.

As much of the content is on phytochemistry, the opinion of a reviewer who is an expert in phytochemistry should be heard.

See some suggestions and corrections in the pdf of the review.

Author Response

Dear Reviewer,

Thank you very much for your kind words and your helpful suggestions. We corrected the points raised by you and added the requested details on the location of collection and the herbarium.

Reviewer 2 Report

Dear Authors,

the manuscript titled "Chemical and Biological Evaluation of Amazonian Medicinal Plant Vouacapoua americana Aubl." describes the isolation and structure elucidation of the compounds from V. americana and their activity on fungi, gram-negative and -positive bacteria as well as on different cancer cell lines. The work may be interested for the readers and shows interesting results in the fields of diterpenoids structure and activity. However, there are some points need explanation/corrections:

1. Abstract - the information about the isolated compounds should be more clear (like from page 4, lines 110-113) and their names should be also indicated.

2. Results concerning the cytotoxicity and microbial assays should be shown for all the tested compounds on all the bacteria strains, fungi and cell lines (graphs, tables). Also, in the case of cytotoxicity results, the positive and negative control also should be indicated.

3. Discussion section - the study on similar compounds and their activity on cancer cells should be mentioned and cited in this section.

4. Methods - methods describing the microbiological and cytotoxicity tests should be shortly described. Also, what kind of solvent was used to dissolve the tested compounds in these experiments? What was the maximal concentration of the solvent added to the bacteria and cells (the negative control)?

Author Response

Dear Authors,

the manuscript titled "Chemical and Biological Evaluation of Amazonian Medicinal Plant Vouacapoua americana Aubl." describes the isolation and structure elucidation of the compounds from V. americana and their activity on fungi, gram-negative and -positive bacteria as well as on different cancer cell lines. The work may be interested for the readers and shows interesting results in the fields of diterpenoids structure and activity. However, there are some points need explanation/corrections:

Dear Reviewer,

Thank you very much for your kind words and your helpful comments. Please find our answers below.

  1. Abstract - the information about the isolated compounds should be more clear (like from page 4, lines 110-113) and their names should be also indicated.

A1) As suggested, we included the names of all isolated compounds into the abstract.

  1. Results concerning the cytotoxicity and microbial assays should be shown for all the tested compounds on all the bacteria strains, fungi and cell lines (graphs, tables). Also, in the case of cytotoxicity results, the positive and negative control also should be indicated.

A2) We added the results of all cytotoxicity and antimicrobial assays to the supporting information (Tables S3 to S5), also indicating positive and negative controls.

  1. Discussion section - the study on similar compounds and their activity on cancer cells should be mentioned and cited in this section.

A3) We added nine references [20–28] for similar compounds with cytotoxic effects to our manuscript and discussed the references also with regard to our structures in section 2.3. (lines 188 to 208).

  1. Methods - methods describing the microbiological and cytotoxicity tests should be shortly described. Also, what kind of solvent was used to dissolve the tested compounds in these experiments? What was the maximal concentration of the solvent added to the bacteria and cells (the negative control)?

A4) Methods describing the microbiological and cytotoxicity tests are included in the materials and methods section. We used DMSO to dissolve the compounds and to prepare the stock solution for bioassays. As shown in section 3.4. and the supporting information, the final DMSO concentration was 0.5% (negative control).

Round 2

Reviewer 2 Report

Dear Authors,

the comments have been take into account. The manuscript is now suitable for publication.